# UNIFIED MULTI-TEACHER DISTILLATION ACROSS HYBRID NEURAL ARCHITECTURES

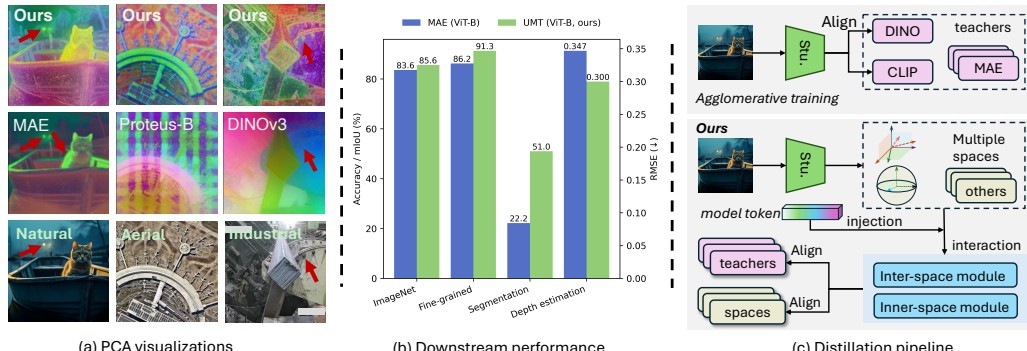

(a) PCA visualizations  (b) Downstream performance  (c) Distillation pipeline

Figure 1: We introduce a unified multi-teacher (UMT) distillation paradigm to integrate general vision knowledge embedded in multiple pre-trained vision foundation models. The PCA feature visualization in (a) shows that the proposed method can extract rich semantics and distinct details across natural, aerial, and industrial images. With knowledge integration from multiple vision foundation models, our method can achieve considerable performance improvements across diverse vision tasks as illustrated in (b). The key to our method is the multi-space interaction mechanism, presented in (c), which enables high-performance multi-teacher distillation using only ImageNet data.

## ABSTRACT

Multi-teacher distillation has recently drawn attention to compress vision knowledge in multiple vision foundation models into a versatile student. The latest multi-teacher distillation techniques for foundation models typically require 1B of training data, rendering a prohibitive training cost in resource-constrained scenarios. Moreover, these methods usually adopt vanilla feature alignment between the student and multiple teachers, which neglects the heterogeneity between teachers, and are therefore highly susceptible to competition and conflicts among teachers that hinders knowledge transfer. To address these limitations, we propose a unified framework to transfer knowledge embedded within multiple vision foundation models into both convolution networks (CNNs) and vision Transformers (ViTs) through training on a 1000× smaller ImageNet-1k dataset. Specifically, we introduce a learnable model token that interacts with visual features across multiple representational spaces. These interactions are mediated through alternating intra-space and inter-space modules, enabling joint feature alignment across diverse source models and architectures. This simple yet effective strategy facilitates unified knowledge transfer from pre-trained Transformers, CNNs, and their combinations—without relying on complex feature-level distillation. Hence, it also establishes an innovative paradigm for cross-architecture distillation. Extensive experiments demonstrate that the resulting model surpasses all its source models in downstream transfer performance, establishing a new sketch for acquiring vision foundation models.

## 1 INTRODUCTION

Vision foundation models (VFMs) (He et al., 2022; Dosovitskiy et al., 2020; Oquab et al.) have revolutionized computer vision by providing powerful, transferable representations that generalize across diverse downstream tasks. These models, including Transformers (Dosovitskiy et al., 2020; Radford et al., 2021) and Convolutional Neural Networks (CNNs) (Liu et al., 2022b), exhibit strong

performance on a spectrum of vision tasks, including classification, segmentation, and detection when trained on large-scale datasets.

Current approaches for acquiring such foundation models often involve large-scale pre-training, making training these models inaccessible for most researchers due to limited computation resources and unpublished datasets (Sun et al., 2017). Moreover, the surrogate pre-training objectives are often misaligned with downstream vision tasks. For instance, CLIP (Radford et al., 2021) trains vision-language foundation models through an image-text contrastive alignment objective, which may facilitate image classification while underperforming on fine-grained classification as well as dense vision tasks (Rao et al., 2022). On the contrary, Mask autoencoders (MAE) (He et al., 2022) excels at extracting distinct details with the help of a patch-level regression target, but in turn lacks in semantic perception ability. To clarify this, we present a PCA feature visualization in Figure 1(a), where MAE tends to capture distinct contents, but cannot easily distinguish between different objects (both the lamp, boat edge, and cats are colored green). The latest method DINOv3 (Siméoni et al., 2025) has become a milestone by scaling up vision model pre-training. However, it tends to compromise distinct details with better semantic understanding as illustrated in Figure 1(a).

Hence, it remains unclear which foundation model should be chosen for a specific downstream vision task. To obtain a robust pre-trained foundation model that exhibits versatile potential at various downstream vision tasks, we consider integrating the knowledge from multiple pre-trained models into a single one. With knowledge inherited from multiple pre-trained models, a single model could be leveraged as a versatile and stronger go-to choice for a majority of downstream tasks. A handy technique would be knowledge distillation (Hinton et al., 2015), where the student model learns by mimicking the behavior of the teacher model on a proxy dataset. However, existing distillation-based approaches such as MobileSAM (Zhang et al., 2023), TinyCLIP (Wu et al., 2023), and Proteus Zhang et al. (2025b) primarily concentrate on transferring the knowledge of a specific vision foundation model into smaller variants, usually suffering from a performance degradation under the single teacher-student distillation setting. This shortcoming renders them incapable of aggregating general vision knowledge within multiple vision foundation models.

Recently, the RADIO series (Ranzinger et al., 2024; Heinrich et al., 2025) develop an agglomerative learning approach to distilling multiple VFMs, *e.g.*, SAM (Kirillov et al., 2023) and DINO-v2 Oquab et al., into a specific student architecture. Albeit successful, this approach requires training on billions of samples (Gadre et al., 2023), rendering the training cost unacceptable under resource-constrained scenarios. Moreover, the vision foundation models are usually pre-trained on different data sources with distinct optimization objectives. Therefore, directly adopting feature alignment between students and multiple teachers like RADIO may suffer from conflicts and competition among multiple teachers Liu et al. (2020), which requires hand-crafted techniques such as multi-resolution training in RADIO-v2.5 to stabilize the learning process. Furthermore, the competition and conflicts among multiple teachers make it nearly impossible to transfer knowledge from multiple teachers when the student presents various architectures. To address these limitations, we consider a more general multi-teacher distillation scenario where multiple heterogeneous teachers present with various architectures or different pre-training objectives, posing a greater challenge for multi-teacher distillation. Given the prohibitive cost of pre-training and the substantial expense of current multi-teacher distillation for foundation models, we raise this critical question in this paper: *Can we develop an economical, performant, and universal approach to aggregating the knowledge of multiple pre-trained vision models into any single student?*

To answer the above question, we develop a unified multi-teacher distillation (UMT) approach that is capable of transferring knowledge across hybrid architectures. UMT aims to explore feature interactions within multiple representational spaces, thereby mitigating the heterogeneous feature alignment issue among different architectures. At its core is the introduction of a learnable model token that is attached to the student's features projected into multiple representational spaces, each of which is responsible for aligning with a specific teacher. Then, through the alternating cross-space and intra-space modules, our model token dynamically interacts with visual features in different representational spaces, consequently enabling effective feature alignment without requiring complex feature-wise matching or manual design of distillation objectives. This simple yet powerful design bypasses many of the constraints in traditional distillation, allowing scalable and architecture-agnostic knowledge fusion. Furthermore, we propose to use the ImageNet-1K as our proxy dataset to perform distillation training, and show that our method achieves better performance than the pioneering method of RADIO (Ranzinger et al., 2024) by using $1000\times$ fewer training samples. Figure 1(b)

demonstrates that the multi-teacher distilled student can dramatically outperform a single pre-trained foundation model across various vision tasks, showcasing the potential and superiority of the proposed approach. Our contributions are summarized as follows:

- We propose a resource-efficient method for transferring knowledge from multiple pre-trained models of heterogeneous architectures, enabling more robust and adaptive vision foundation models for a wide range of tasks.

- We introduce a learnable model token that bridges representational gaps across different architectures by interacting with their features in a structured and modular way. A cross-space and intra-space interaction mechanism further enhances this interaction.

- Our framework provides a unified and extensible approach to multi-teacher distillation, with promising implications for broader areas such as cross-modal knowledge transfer and multi-teacher learning.

## 2 RELATED WORK

**Vision foundation models**. Vision foundation models (VFMs) have quickly reshaped the landscape of computer vision tasks, powered by advances in large-scale pre-training strategies. Early contrastive learning methods such as MoCo (He et al., 2020), SimCLR (Chen et al., 2020), and DINO (Caron et al., 2021) learn discriminative features by contrasting positive and negative image pairs, with DINO notably using a self-distillation setup to train Vision Transformers (ViTs) without labels. In parallel, masked image modeling approaches like MAE (He et al., 2022) and BEiT (Bao et al., 2021) draw inspiration from BERT (Devlin et al., 2019), masking large portions of the input and training the model to reconstruct them, thereby encouraging global context understanding. Beyond single-modality learning, multi-modal pre-training strategies such as CLIP (Radford et al., 2021) and ALIGN (Jia et al., 2021) use contrastive learning between image-text pairs to align representations across modalities, enabling zero-shot generalization and semantic reasoning. Similarly, Segment Anything (SAM) (Kirillov et al., 2023) leverages large-scale segmentation data to build a universal visual backbone. The latest self-supervised approach DINO-v3 (Siméoni et al., 2025) successfully scales up VFMs with careful dataset curation and advanced training techniques. Despite their shared goal of general-purpose representation learning, these VFMs differ significantly in architecture (*e.g.*, CNNs vs. Transformers), objective (*e.g.*, contrastive vs. reconstruction), and task preference—raising the question of how to integrate their complementary strengths into a single, versatile model.

**Knowledge distillation**. Knowledge distillation (Hinton et al., 2015) has been widely used to transfer knowledge from a single large teacher to a smaller student. Traditional methods work by matching the intermediate features (Romero et al., 2015) or model predictions (Hinton et al., 2015) between the teacher and the student, and often assume identical architectures (*e.g.*, ViT-to-ViT or ResNet-to-ResNet). Recent methods such as MobileSAM (Zhang et al., 2023), TinyCLIP (Wu et al., 2023), and Proteus (Zhang et al., 2025b) continue this trend, typically involving one-to-one teacher-student pairs trained on large-scale datasets. However, this formulation fails to address settings with multiple, heterogeneous teachers—such as a hybrid set of pre-trained Transformers and CNNs teachers—each offering complementary knowledge and strengths. Existing methods that attempt cross-architecture feature alignment require careful manual design and struggle to scale as the number of teachers increases (Hao et al., 2023; Zhang et al., 2025a; Liu et al., 2022a). In contrast, our work introduces a unified and architecture-agnostic solution for fusing knowledge from diverse VFMs into a single student model, enabling robust generalization without hand-crafted feature matching.

**Mutli-teacher distillation for VFMs**. (Ranzinger et al., 2024) introduce an agglomerative distillation approach to reducing the domain knowledge embedded in various vision foundation models into a single student model. Through a direct global-level and patch-level alignment with CLIP (Radford et al., 2021), SAM (Kirillov et al., 2023), and DINO-v2 (Oquab et al., 2023), the consequent student model is shown to inherit rich knowledge for various vision-related tasks. Soon, RADIO-v2.5 (Heinrich et al., 2025) extends this concept with multi-resolution distillation, which facilitates high-resolution vision tasks. However, these approaches directly employ feature alignment between student and teachers, which risking unsatisfactory optimization with heterogeneous features. Moreover, they require training on the large-scale DataComp-1B dataset, incurring enormous computation-source demand. On the contrary, we opt for a more economical multi-teacher distillation approach, regardless of the architectures as well as surrogate pre-training objectives.

Figure 2: A schematic illustration of the proposed universal multi-teacher (UMT) distillation framework. UMT transfers knowledge from multiple teachers (CNN, ViT, or a hybrid set of both)) to a student model (either CNN or ViT). A learnable model token interacts with features of the student within multiple teacher-specific representational spaces. After processing alternatively by the proposed cross-space and intra-space module $N$ times, the model token is required to align with the global-level feature of teachers. Additionally, the visual features from student are aligned with teachers correspondingly.

## 3 METHODOLOGY

In this section, we formally formulate the multi-teacher knowledge distillation problem, alongside the challenges. We then present detailed descriptions of our unified multi-teacher distillation (UMT) framework and its components, including feature re-projection and multi-space token interaction.

### 3.1 UNIFIED MULTI-TEACHER DISTILLATION OVERVIEW

We consider a multi-teacher knowledge distillation set-up where we have access to $M$ pre-trained vision foundation models as teachers, denoted $\mathcal{T} = \{T_i\}_{i=1}^M$. Each of the teacher models may belong to different vision architecture families (*e.g.*, CNN, ViT) and have been trained on data of different sources (*e.g.*, ImageNet-1K, LAION-2B). The objective is to train a single student model $S$, (*e.g.*, CNN or ViT) that maximally inherits the diverse knowledge from all teachers. To achieve this goal most economically, unlike the pioneering RADIO (Heinrich et al., 2025) that requires billions of samples (Gadre et al., 2023) to train, we instead employ ImageNet-1K—with $1000\times$ fewer images—as our proxy dataset $\mathcal{D}$ to align the student's representations with those from all teachers. Nonetheless, naive one-to-many representation alignment may suffer from competitions or even conflicts among multiple teachers' representations, leading to compromised knowledge learned by the student (Yuan et al., 2021; Liu et al., 2020).

To reconcile among multiple, possibly heterogeneous or conflicting teacher features, we propose to project the student's feature into $M$ different representational spaces, each corresponding to and responsible for assimilating the vision knowledge embedded in one of the $M$ teachers. Unlike existing approaches (Ranzinger et al., 2024; Heinrich et al., 2025) that enforce naive one-to-many feature alignment that is prone to conflicts, we introduce a novel learnable model token $P_m \in \mathbb{R}^{d_m}$ to interact with the student's feature in respective representational spaces as illustrated in Figure 2. Through extensive interaction with student's features in $M$ representational spaces, the model token becomes capable of capturing model-agnostic general vision knowledge across all teachers. Hence, the model token could further promote the alignment between the student and $M$ teachers. To facilitate this process, we compel the model token to be aligned with all teachers' global-level image representations as shown in Figure 2. Hence, the student visual features, after interaction with the model token within multiple representational spaces, could easily be aligned with the features obtained from all teachers.

### 3.2 FEATURE RE-PROJECTION INTO REPRESENTATIONAL SPACES

As illustrated in Figure 2, for a given image $x$ sampled from proxy dataset $\mathcal{D}$, the student model $S$ outputs visual feature $F_S \in \mathbb{R}^{H \times W \times d_S}$ if it adopts a ConvNet architecture, where $H$, $W$, and $d_S$ denote its height, width, and channel dimension, or $F_S \in \mathbb{R}^{(1+L) \times d_S}$ if a ViT architecture, where $L$ is the length of the patch sequence and "1" corresponds to the specific class token. Analogously,

each teacher $T_i$ processes the same input image $x$ and produces visual feature $F_{T_i} \in \mathbb{R}^{H \times W \times d_{T_i}}$ or $F_{T_i} \in \mathbb{R}^{(1+L) \times d_{T_i}}$, depending on whether it adopts a ConvNet or a ViT architecture.

Knowledge distillation assumes that the student $S$ learns from a specific teacher $T_i$ if $F_S$ is imposed to align with $F_{T_i}$ for all samples in the proxy dataset. Therefore, if $F_S$ could be well aligned with visual features $\{F_{T_i}\}$ from all teachers $\{T_i\}$, then we can fairly assume that the student has assimilated their knowledge (Romero et al., 2015). Since the features from multiple teachers present in distinct representational spaces due to different architectures, various pre-training data sources, and specific optimization objectives, we first need to re-project $F_S$ into their respective representational spaces, each responsible for alignment with a specific teacher. As such, a total of $M$ projection operations are involved, denoted by $\Gamma_i,\ i = 1, 2, \cdots, M$, for the set of $M$ teachers.

$$F_S^i = \Gamma_i(F_S),\ i = 1, 2, \cdots, M, \tag{1}$$

where $F_S^i \in \mathbb{R}^{H \times W \times d_{T_i}}$ or $F_S^i \in \mathbb{R}^{(1+L) \times d_{T_i}}$.

### 3.3 FEATURE INTERACTION IN MULTIPLE REPRESENTATIONAL SPACES

Conventional feature-based distillation transfers the teacher's capability to the student by aligning their intermediate features (Romero et al., 2015). However, in our multi-teacher distillation setup, blindly forcing each $F_S^i$ to match $F_{T_i}$ is sub-optimal for at least two reasons: First, not all teachers are pre-trained on the proxy dataset, hence a strict alignment between student and teacher will suffer from dataset bias (Zhang et al., 2025b), hindering student learning. Second, the representational spaces of different teachers can be highly diverse. For instance, an MAE pre-trained model projects images into an Euclidean space with the mean square error (MSE) criterion, while a contrastive learning-based model embeds images into a hypersphere (Wang & Isola, 2020). Therefore, designing proper alignment criteria that takes account of these discrepancies between multi-teacher representations becomes vital. To further enhance the adaptation capability of distilled model, the interaction among representational spaces maybe helpful. Since visual features of teachers are represented in different spaces, considering their interaction could boost the knowledge transfer of general visual representation.

In this paper, we introduce a learnable model token $\mathrm{P}_m \in \mathbb{R}^{d_m}$, and further develop cross-space and intra-space interaction mechanisms to facilitate alignment issue among all teachers, while simultaneously promoting the knowledge transfer in various representational spaces.

**Cross-space interaction**. As illustrated in Figure 2, we first expand the model token $\mathrm{P}_m$ to match the shape of each projected features $F_S^i$ and get them concatenated, before sending to a cascade of alternating intra-space and cross-space interaction modules. With model token incorporated to all representational spaces, we can safely explore their interactions. Concretely, let $\psi_C^j$ denote the $j$-th cross-space interaction module, we have

$$F_C^j = \psi_C^j([\{F_S^i, \mathrm{P}_m\}]),\ \text{if}\ j = 1, \tag{2}$$

$$F_C^j = \psi_C^j([F_{I,i}^{j-1}]),\ j \geq 2, \tag{3}$$

where $[\cdot]$ denotes the concatenation operation, and $F_{I,i}^{j-1}$ denotes the output feature of $(j-1)$-th intra-space interaction module corresponding to the $i$-th representational space.

It is noteworthy that the concatenation results in a total number of $M \times (d_T + d_m)$ channels, which scales linearly with the number of teachers. Naively processing the concatenated features with a fully channel-wise convolution or attention would incur unacceptable computation cost. To reduce the computational complexity, we adopt following techniques:

- For CNN students, inspired by Liu et al. (2022b), we construct the module with two basic convolutional layers, the depth-wise convolution and $1 \times 1$ point-wise group convolution.
- For Transformer students, we enlarge the head number linearly to the number of teachers.

**Intra-space interaction**. Each representational space is dedicated to learning from a specific teacher, embedding images into a meaningful and structured form. The cross-space interaction module helps aggregate visual information from all spaces. To ensure that this visual information is accurately represented within each space, it is essential to carefully reorganize the features accordingly. For this

Table 1: A comprehensive comparison across ImageNet classification, fine-grained classification, image segmentation, and depth estimation. † denotes methods for which official results are reported. The best distillation results are in **bold**.

| Method | Student Arch. | Setting | # imgs | ImageNet | Fine-grained Cls. | | | Segmentation (Probing) ADE20K (mIoU↑) | Depth Estimation NYUv2 (RMSE↓) |
| --- | --- | --- | --- | --- | --- | --- | --- | --- | --- |
| | | | | | Aircraft | Caltech101 | CUB | | |
| Classification (He et al., 2016) | | - | 1.2M | 80.9 | 81.6 | 90.5 | 83.3 | 12.4 | 0.439 |
| MAE (He et al., 2022) | ResNet-50 | - | 1.2M | 65.0 | 83.6 | 88.0 | 71.7 | 15.83 | 0.428 |
| DINO (Caron et al., 2021) | | - | 1.2M | 80.0 | 76.1 | 89.6 | 74.8 | 18.4 | 0.426 |
| UMT (**ours**) | | CNN→CNN | 1.2M | 78.0 | 85.4 | 89.9 | 79.7 | 20.5 | 0.408 |
| Classification (Liu et al., 2022b) | | - | 1.2M | 83.8 | 87.0 | 92.9 | 90.6 | 18.0 | 0.355 |
| Spark (Tian et al., 2023) | | - | 1.2M | 84.8 | 92.1 | 91.6 | 82.6 | 31.3 | 0.329 |
| DINOv3 (Siméoni et al., 2025) | ConvNeXt-B | - | 1689M | 86.3 | 94.0 | 92.7 | 89.1 | 36.4 | 0.293 |
| UMT (**ours**) | | CNN→CNN | 1.2M | 83.0 | 88.7 | 93.1 | 88.7 | 34.6 | 0.361 |
| UMT (**ours**) | | ViT→CNN | 1.2M | 84.1 | 90.0 | 93.5 | 88.7 | 34.7 | 0.349 |
| CLIP (Radford et al., 2021) | ViT-B/16 | - | 400M | 84.2 | 82.1 | 90.7 | 85.1 | - | - |
| MAE (He et al., 2022) | ViT-B/16 | - | 1.2M | 83.6 | 85.9 | 91.6 | 81.2 | 22.2 | 0.347 |
| DINOv3 (Siméoni et al., 2025) | ViT-B/16 | - | 1689M | 86.2 | 92.6 | 93.5 | 90.3 | 50.9 | 0.293 |
| Proteus† (Zhang et al., 2023) | ViT-B/14 | ViT→ViT | 1.2M | 84.9 | - | - | - | - | 0.304 |
| RADIOv2.5† (Heinrich et al., 2025) | RADIOv2.5-B | ViT→RADIO | ~1400M | - | - | - | - | **48.9** | - |
| UMT (**ours**) | ViT-B/16 | ViT→ViT | 1.2M | **85.6** | 90.3 | 94.3 | 89.2 | 46.5 | **0.300** |
| CLIP (Radford et al., 2021) | ViT-L/14 | - | 400M | 86.7 | 84.4 | 91.4 | 88.2 | - | - |
| MAE (He et al., 2022) | ViT-L/16 | - | 1.2M | 85.9 | 91.5 | 92.4 | 85.5 | 27.1 | 0.267 |
| DINOv3 (Siméoni et al., 2025) | ViT-L/16 | - | 1689M | 88.4 | 94.5 | 93.7 | 91.5 | 54.6 | 0.224 |
| Proteus† (Zhang et al., 2023) | ViT-L/14 | ViT→ViT | 1.2M | 86.2 | - | - | - | - | **0.240** |
| RADIOv2.5† (Heinrich et al., 2025) | RADIOv2.5-L | ViT→RADIO | ~1400M | - | - | - | - | 51.5 | - |
| UMT (**ours**) | ViT-L/16 | ViT→ViT | 1.2M | **87.5** | 92.1 | 93.8 | 91.4 | 50.3 | 0.258 |

purpose, we design an intra-space interaction mechanism as illustrated in Figure 2, placed after the cross-space interaction module. As a result, the intra-space module operates independently on each projected features. Denote $F_{C,i}^{j}$ be the output corresponding to the $i$-th teacher, chunked from the $j$-th cross-space module. And let $\phi_I^j$ be the intra-space module, then we have $F_{I,i}^{j+1} = \phi_I^j(F_{C,i}^j)$ as the reorganized feature for the $i$-th teacher.

## 3.4 Multi-teacher Feature Alignment

Current multi-teacher distillation approaches like Zhang et al. (2025b) and Heinrich et al. (2025) harshly align the student features with different teachers at multiple levels, risking in representation collision among teachers. For instance, forcibly aligning the global `cls` token with both CLIP and MAE simultaneously may in turn harm the global visual perception of students. Hence we argue that a careful alignment between student and teachers must be taken into consideration.

**Model token alignment**. We opt for the model token to perform a global-level alignment with teachers. With rich interaction with visual features in multiple representational spaces, this alignment can in turn force each representational space characterizing the teacher's feature property. With this goal, we first pool the ultimate feature $F_{I,i}^N$, and chunk the model token part to obtain a global-level token in the $i$-th representational space, corresponding for aligning with the global-level token of the $i$-th teacher. Specifically, we pool the CNN's ultimate feature and use the `cls` token of a ViT as supervision, respectively. For simplicity, we denote the model token alignment objective as $\mathcal{L}_{teacher}^m$ corresponding for a specific teacher.

**Student feature alignment**. As shown in Figure 2, we chunk the student feature part from $F_{I,i}^N$ to align student feature with each teacher. It is noteworthy that with different pre-training objectives, the ultimate features lie in diverse manifolds. As a result, we apply different feature constraints for different teachers. For teachers excelling at extract global-level or semantic clues, *e.g.*, CLIP, DINO, and classification-pre-train, we employ a cosine similarity constraint. While for reconstruction-based pre-training models such as MAE, we use Mean Square Error (MSE) loss. For simplicity, we refer to the feature alignment loss as $\mathcal{L}_{teacher}^f$ for a specific teacher.

With the aforementioned feature interaction mechanism and multi-teacher alignment objective, we could distill multiple teachers into a single student from scratch on the ImageNet dataset. Details of the model token alignment and student feature alignment losses are provided in Appendix A.3.

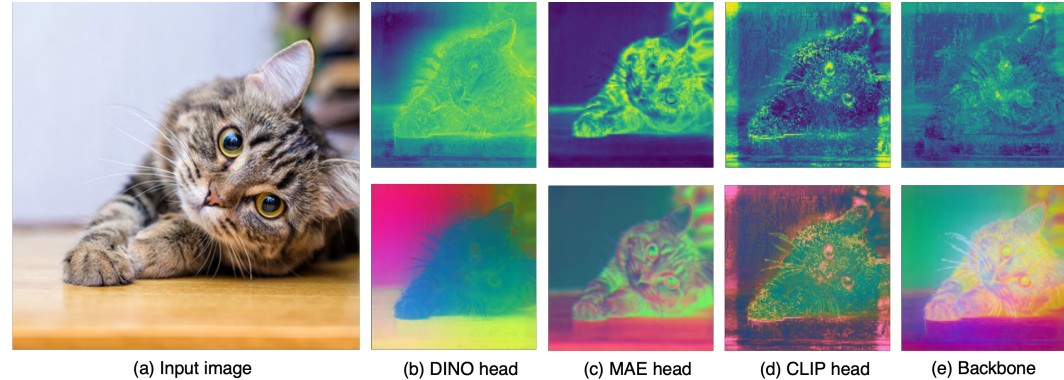

(a) Input image        (b) DINO head     (c) MAE head     (d) CLIP head     (e) Backbone

Figure 3: Cosine similarity (top) and PCA (bottom) visualization of deep features at different places within the ViT-B/16 student. The input image resolution is $2048 \times 2048$, resulting in $128 \times 128$ spatial features. **Top**: cosine similarity map between class token and patch tokens. **Bottom**: PCA visualization by projecting features into RGB space.

## 4 EXPERIMENTS

### 4.1 EXPERIMENTAL SETUP

**Implementation details**. We conduct multi-teacher distillation on various student architectures, including ResNet-50 (He et al., 2016), ConvNeXt-B (Liu et al., 2022b), ViT-B/16 (Dosovitskiy et al., 2020), and ViT-L/16 (Dosovitskiy et al., 2020). For each randomly initialized student, we employ its pretrained classification-based (CLIP for ViT), MAE-based, and DINO-based counterparts as teachers. For DINO-based teachers, we adopt the latest DINO-v3 ConvNeXt-B and ViT series. Distillation training is conducted on the ImageNet-1K dataset following a standard 300-epoch training configuration with $224 \times 224$ image resolution. We use two cross-space and intra-space modules for CNN student, while one for ViT student, respectively. More details about the training configurations and the cross-space and intra-space module designs are provided in Appendix A.2.

**Benchmarks**. We evaluate the model's capability across a wide array of vision tasks and datasets: image classification on ImageNet-1K (Deng et al., 2009), fine-grained image classification on Aircraft (Maji et al., 2013), Caltech101 (Fei-Fei et al., 2007), and CUB2011 (Wah et al., 2011) datasets, semantic segmentation on ADE20K (Zhou et al., 2017), and depth estimation on NYUv2 (Silberman et al., 2012). More details can be found in the Section A.1.

### 4.2 MAIN RESULTS

We examine the capability of the UMT-distilled student across various downstream vision tasks by comparing their performance with corresponding teachers. We select two recent advanced distillation approaches, Proteus (Zhang et al., 2025b) and RADIO-v2.5 (Heinrich et al., 2025) for comparison. In particular, we collect the officially reported metrics for these two methods for a fair comparison.

**Multi-teacher distillation into Transformers**. We distill the official CLIP ViT-B/16, MAE-pretrained ViT-B/16, and DINO-v3 ViT-B/16 models into a vanilla ViT-B/16 student. We do not adopt the advanced 2D rope position embedding (Su et al., 2024), hybrid architecture design like Heinrich et al. (2025), and register tokens Darcet et al. (2024) that could benefit the student's performance. Table 1 presents quantitative results across four fundamental vision tasks.

We observe that the distilled student achieves competitive performance on ImageNet and fine-grained classification. Notably, on certain tasks the student even surpasses teacher, such as on Caltech101 where it outperforms all three teachers. On dense prediction tasks, the ViT-B/16 student consistently outperforms the MAE teacher with a 26.7% linear probing mIoU gain on semantic segmentation and a 0.047 RMSE improvement on depth estimation. The distilled student also outperforms Proteus-B on both ImageNet classification and depth estimation. On the other hand, since the distilled students tends to provide features with distinct details, linear probing may not unleash its potential in dense

Table 2: Multi-teacher distillation using teachers of different architectures. We use different combinations of teacher models to distill a ConvNeXt-B student. The best results are in **bold**.

| Combination | Teacher-I (CLS) | Teacher-II (MAE) | Teacher-III (DINO) | $\mathcal{L}_{CLS}^m + \mathcal{L}_{MAE}^m + \mathcal{L}_{DINO}^m$ | $\mathcal{L}_{CLS}^f + \mathcal{L}_{MAE}^f + \mathcal{L}_{DINO}^f$ | $\mathcal{L}_{DINO}^m + \mathcal{L}_{DINO}^f$ |
|---|---|---|---|---|---|---|
| Homogeneous | ConvNeXt-B | ConvNeXt-B | ConvNeXt-B | 0.20 | 0.23 | 0.31 |
| Hybrid | CLIP | ConvNeXt-B | ConvNeXt-B | 0.11 | 0.27 | 0.26 |
| Hybrid | ConvNeXt-B | ViT-B/16 | ConvNeXt-B | 0.18 | 0.48 | 0.60 |
| Hybrid | CLIP | ViT-B/16 | ViT-B/16 | **0.09** | **0.08** | **0.10** |

segmentation which requires continuous feature maps. Nonetheless, with vision knowledge integrated from other teachers, the student even outperforms DINO-v3 on the Caltech101 dataset. Note that our method only falls slightly short of RADIO-v2.5 on semantic segmentation, where they rely on explicit segmentation knowledge transfer from SAM (Kirillov et al., 2023) on a $1000\times$ larger dataset.

**Multi-teacher distillation into CNNs**. We further adopt UMT to distill two CNN variants, i.e., ResNet-50 and ConvNeXt-B, where their ImageNet classification pre-trained, MAE, and DINO counterparts are used as teachers. Quantitative results are presented in Table 1. We observe that the ResNet-50 student achieves substantial improvements on dense prediction tasks, including segmentation and depth estimation, whereas the ConvNeXt-B student struggles to integrate knowledge from all teachers as shown in Table 6. We conjecture that the strong inductive bias of CNN with dataset bias will amplify the difficulty for multi-teacher distillation among CNNs. We will show that hybrid multi-teacher distillation can alleviate this issue.

**Multi-teacher distillation from hybrid teacher architectures**. Cross-architecture distillation remains an open challenge due to difficulties in reconciling heterogeneous features (Hao et al., 2023; Zhang et al., 2025a). However, as illustrated in Table 1, distilling a ConvNeXt-B student using all *ViT* teachers consistently outperforms the homogeneous distilled student across four vision tasks. Here, in contrast to the standard single teacher-student cross-architecture distillation, we make a counter-intuitive observation: *in the presence of dataset bias among the student and teachers, using all ViT teachers can maximally alleviate the performance degradation when the student is CNN*. We explore this by fixing the student to ConvNeXt-B, and change the architecture combination of all teachers. As shown in Table 2, by substituting a classification-pretrained CNN teacher with CLIP, we can obtain a slight training loss decrease. However, if we simultaneously employ a ViT-based MAE teacher and a CNN-based DINO teacher, the multi-teacher distillation becomes challenging due to the heterogeneous spatial feature alignment. Surprisingly, with all ViT teachers, the multi-teacher distillation becomes much easier. One step further, we observe that the DINO loss accounts for the performance variation as shown in Table 2, caused the dataset bias between LVD-1689M and ImageNet-1K. Therefore, we make this safe assertion that *all ViT teachers can alleviate the dataset bias influence for multi-teacher distillation with a CNN student*.

### 4.3 ABLATION STUDIES

In this section, we conduct experiments to verify the importance of the introduced model token, together with the cross-space and intra-space modules. We also show that the student distilled using the proposed approach can extract robust features against various input resolutions. Besides, we examine the impact of feature alignment objectives, which underscores the importance of careful alignment between the student and teachers.

**Role of the model token**. We first investigate the role of the model token. We distill a ConvNeXt-B student by eliminating the model token. The results in Table 3 suggest that model token helps optimize the feature alignment loss. With the interaction between the model token and the student visual features within multiple representational spaces, the model-agnostic general vision knowledge captured by the model token facilitates the alignment between student and all teachers.

Table 3: Ablation on model token and the cross-space and intra-space modules.

| Configuration | Model token loss | Feature loss |
|---|---|---|
| w/o model token | - | 0.29 |
| w/o cross-space module | 0.18 | 0.35 |
| w/o intra-space module | 3.56 | 2.60 |
| UMT (ours) | 0.20 | 0.23 |

**Effectiveness of intra- and cross-space modules**. We distill a ConvNeXt-B student by ablating the intra-space and cross-space modules to understand their effectiveness. As shown in Table 3, the feature re-organization by the intra-space module is crucial for both model token and student

feature alignments. Additionally, the cross-space module could help reduce the feature alignment loss between student and teachers.

**Effect of multi-space feature interaction**. We examine the benefits of using the features after interaction facilitated by the model token, compared to using the raw features from the backbone. We conduct fine-grained classification on the CUB2011 dataset, with ResNet-50 as the student backbone. We adopt the model token projected to DINO, MAE, and classification head for classification. From Table 4, using the features from backbone yields unsatisfactory performance, with a 5.9% accuracy degradation.

Table 4: Effect of using different features for classification on the CUB2011 dataset.

| Backbone | DINO head | MAE head | CLS head |
|---|---|---|---|
| 73.8 | 77.9 (+4.1) | 77.0 (+3.2) | 79.7 (+5.9) |

With the model token being projected into different heads (DINO, MAE, CLS), we observe a consistent accuracy gain. Additionally, we present a PCA visualization analysis in Figure 3, where the projected DINO head perceives a better sematic understanding as well as exhibits powerful detail capturing capability. We leave detail discussion in Appendix A.6.

**Choices of alignment objectives**. We evaluate the performance of the ViT-L/16 student trained using different DINO objective on three fine-grained datasets. As presented in Table 5, using a cosine similarity constraint could better transfer the semantic understanding ability to the student.

Table 5: Impact of different feature alignment objectives.

| DINO objective | Aircraft | Caltech101 | CUB2011 | Average |
|---|---|---|---|---|
| MSE | 89.4 | 92.8 | 90.6 | 90.9 |
| Cosine | 92.1 | 93.8 | 91.4 | 92.4 |

**CKA visualization**. We also adopt the Central Kernel Alignment (CKA) (Kornblith et al., 2019) to quantify the representation similarity between student and all teachers. As shown in Table 6, the student shares a highest CKA metric with MAE since they are all trained on the same dataset. With a larger training dataset scale, where potential dataset bias becomes more prominent, the CKA similarity between student and teacher decreases, resulting in performance drop compared to DINO-v3 teacher as shown in Table 1.

Table 6: CKA feature similarity analysis for ViT-B and ViT-L students.

| Student ImageNet-1K | Level - | CLIP CLIP-400M | MAE ImageNet-1K | DINO-v3 LVD-1689M |
|---|---|---|---|---|
| ViT-B/16 | global | 0.83 | 0.98 | 0.76 |
| | patch | - | 0.95 | 0.81 |
| ViT-L/16 | global | 0.71 | 0.85 | 0.66 |
| | patch | - | 0.92 | 0.73 |

## 5 LIMITATIONS

As can be seen in Table 1, the major limitation lies in the dataset bias incurred between DINO-v3's LVD-1689M dataset and the distillation proxy dataset ImageNet-1K. Specifically, the LVD-1689M dataset contains high-quality images from the Web, public ImageNet dataset, and street-level sequences, covering all visual concepts appearing on the web. However, most images within the ImageNet-1K dataset only contain one major subject, limiting the knowledge transfer of multi-object perception capability from DINO-v3 pre-trained models. Furthermore, the strong inductive bias learned from different datasets poses a challenge in multi-teacher distillation with CNN architectures. Although we find that utilizing all ViT is a viable way to mitigate the dataset bias, it requires further exploration into combating the dataset bias in multi-teacher distillation.

## 6 CONCLUSION

In this paper, we introduce a unified multi-teacher distillation approach to integrate multiple vision foundation models' knowledge into a singe model. The distilled model has shown impressive performance across a range of downstream vision tasks from classification and semantic segmentation to depth estimation. In particular, we introduce a learnable model token, which interacts with visual features at multiple representational spaces, facilitating the heterogeneous feature alignment with different architectures. This powerful strategy paves an economical way to obtain a versatile vision model by training on ImageNet-1K, without requirement for large-scale data sources or complex training procedures.

## 7 ETHICS STATEMENT

This work focuses on developing a resource-efficient framework for multi-teacher distillation of vision foundation models. By reducing the reliance on billion-scale datasets and extensive computational resources, our approach lowers the environmental footprint of training and makes advanced vision models more accessible to researchers with limited resources. This aligns with broader goals of sustainability and inclusiveness in AI research.

## 8 REPRODUCIBILITY STATEMENT

We use the publicly-accessible datasets ImageNet-1k, FGVC-aircraft, Caltech101, CUB2011, ADE20K, and NYUv2. We have elaborated our method and experimental settings in detail in the paper. Additionally, our downstream task training is based on the public MMSegmentation toolkit and follows the public repositories. We will upload our codes in the further for re-implementation of the proposed approach.

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

# A  APPENDIX

## A.1  IMPLEMENTATION DETAILS

We conduct all experiments with 8 NVidia A800 GPUs using the PyTorch (Paszke, 2019) framework. In this section, we provide more details on the training of distillation and downstream benchmarks.

**Multi-teacher Distillation Setting**. We train all students on the ImageNet-1K from scratch with image resolution be $224 \times 224$. The training batch size for ResNet-50, ConvNeXt-B, and ViT-B is set to 1024. For ViT-L experiments, we set the batch size to 512 due to GPU memory constraint. Besides, we employ the mixed precision training to save GPU memory and accelerate training. We freeze all teachers during distillation, which forms a standard and default choice. The advanced AdamW Loshchilov & Hutter (2017) optimizer is employed with a cosine annealing learning rate schedule. The base learning rate is set as 1e-4, with Below we provide details on the student and teachers:

- ResNet-50 student. We utilize the official ImageNet-1K classification pretrained, MAE pre-trained by SparK (Tian et al., 2023), and DINO pre-trained ResNet-50 models as teachers.

- ConvNeXt-B student. We adopt the official ImageNet-1K classification pre-trained, MAE pre-trained by SparK (Tian et al., 2023), and latest DINO-v3 ConvNeXt-B models as teachers.

- ViT-B student. We implement a vanilla ViT-B/16 under the timm framework without register tokens and only using learnable position embedding. We employ the official CLIP ViT-B/16, MAE pre-trained ViT-B/16, along with latest DINO-v3 ViT-B/16 as teachers. It is noteworthy that DINO-v3 employs advanced 2D ROPE position embeddings for better performance.

- ViT-L student. We implement a vanilla ViT-L/16 student using the official timm library. As for teachers, we adopt the official CLIP ViT-L/14, MAE pre-trained ViT-L/16, together with latest DINO-v3 ViT-L/16 models.

**Benchmark Training**. We conduct performance evaluation of the distilled student on four fundamental vision tasks, ImageNet classification, fine-grained classification, image segmentation, and depth estimation.

We first fine-tune all listed models in Table 1 except for Proteus and RADIO-v2.5 on the ImageNet-1K dataset for 10 epochs with a learning rate 1e-4. The training starts with one epoch warmup. We set the batch size as 512. The training input resolution is $224 \times 224$. We adopt a layer-wise learning-rate decay of 0.8 for all models. Once tuning finished for the models, we report their top-1 accuracy on the validation set of ImageNet-1K.

For the dense segmentation on ADE20K dataset, we adopt a linear decoder head following Heinrich et al. (2025) while freezing the backbones. Typically, we adopt a standard 80k training schedule with a training batch size 16 utilizing the off-the-shelf MMSegmentation toolkit. We report the mean Intersection over Union (mIoU) metric on the validation set. The training resolution is fixed as $512 \times 512$ for all models. To enabling training on this resolution, we adopt a bicubic position embedding strategy.

Finally, we conduct downstream depth estimation on the widely adopted NYUv2 benchmark. Following Zhang et al. (2025b), we use the DPT (Ranftl et al., 2021) as the decoder head. Specifically, the DPT decoder head requires hierarchical spatial features together with a summary token as inputs. Therefore, with ResNet-50 and ConvNeXt-B backbones, we utilize the hierarchical spatial features within four stages, and spatially pool these features to obtain a summary token. This design makes the DPT decoder head suitable for depth estimation with CNN models. As for the ViT-B backbone, we form the hierarchical deep features by extracting features at indices $3, 5, 7, 11$. Specifically, the corresponding `cls` tokens are regarded as the summary tokens required by the DPT deocder head. Additionally, the output indices for the ViT-L model is changed to $7, 11, 15, 23$. In practice, We train the student and teachers (except CLIP) on NYUv2 with batch size 16, experiencing a standard 25k training schedule. Following Zhang et al. (2025b), the learning rate for the backbone is scaled by a factor of 0.1. Specifically, we adopt the Root Mean Square Error (RMSE) metric for evaluation.

Table 7: Alleviate the impact of dataset bias by choosing teachers (Student as ConvNeXt-B).

| Teacher-I | Teacher-II | Teacher-III | ADE20K | NYUv2 (RMSE) |
|---|---|---|---|---|
| CLIP (ViT-B/16) | MAE (ViT-B/16) | DINO-v3 (ViT-B/16) | 34.7 | 0.349 |
| DINO-v3 (ConvNeXt-B) | DINO-v3 (ViT-B/16) | - | 35.1 | 0.332 |

## A.2 Cross-space and Intra-space Module Design

**CNN as student**. We adopt two cross-space and intra-space modules for CNN student. The cross-space module is comprised of layer normalization, followed by a depth convolution with $3 \times 3$ kernels, GELU activation, and a $1 \times 1$ convolution layer with $G$ groups. We set $G$ for ResNet-50 to 128 and 64 for ConvNeXt-B.

The intra-space module follows the same architecture design with cross-space module. Additionally, the intra-space module is shared across different representational spaces. We adopt a residual connection to both types of modules.

**ViT as student**. We build the cross-space and intra-space module on the vanilla attention block, featuring a norm-attention-norm-ffn design. Both types of modules share the same architecture. For the cross-space module, we enlarge the head number in the attention block linearly to the number of teachers. Additionally, the intra-space module is shared across multiple representational spaces.

## A.3 Details on Hybrid Architecture Distillation

**Model token alignment loss**. After the alternating cross-space and intra-space modules, we chunk the model token from ultimate deep features at different representational spaces. We the use a shared projection head (LayerNorm-Linear) to align its channels with all teachers. We then pool the model token to align with the teachers' global-level token. For CNN teachers, we utilize their pooling features as supervision. On the contrary, we adopt the `cls` token from ViT teachers except CLIP for supervision. For CLIP teacher, we directly use its image encoding vector as aligning target. We simply adopt MSE loss for all types of teachers.

**Student Feature Alignment Loss**. After the alternating cross-space and intra-space modules, we chunk the student feature from the ultimate deep features. For classification-based and CLIP teachers, we pool the features and utilize a cosine loss for semantic alignment. For MAE teacher, we adopt a MSE loss for patch-level alignment. As for DINO teacher, we utilize a patch-level cosine constraint for the alignment.

## A.4 Robustness to Different Input Resolutions

Heinrich et al. (2025) observes that the student's feature tends to diverge when input resolution increases to 1024. However, since all teachers in our design are pre-trained at similar resolutions, the ultimate student can extract robust features when input resolution changes from $512^2$ to $4096^2$ as illuatrated in Figure 4.

## A.5 Mitigate Dataset Bias Impact by Choosing Teachers

As illustrated in Section 4.2, we can mitigate the influence of dataset bias among teacher and students by substituting all CNN teachers with ViTs. In this section, we observe that reducing the dataset bias between teachers could also improve distillation performance. As shown in table 7, we adopt a DINO-v3 CNN and ViT teacher to distill a ConvNeXt-B student. Since the teachers are all pre-trained on the same LVD-1689M dataest, the dataset bias between them can be neglected. As a result, we obtain a better segmentation and depth estimation performance as shown in table 7.

## A.6 Effect of using different features for classification on CUB2011 dataset.

We continue our discussion about the multi-space feature interaction by analyzing the feature quality of a distilled ViT-B/16 student.

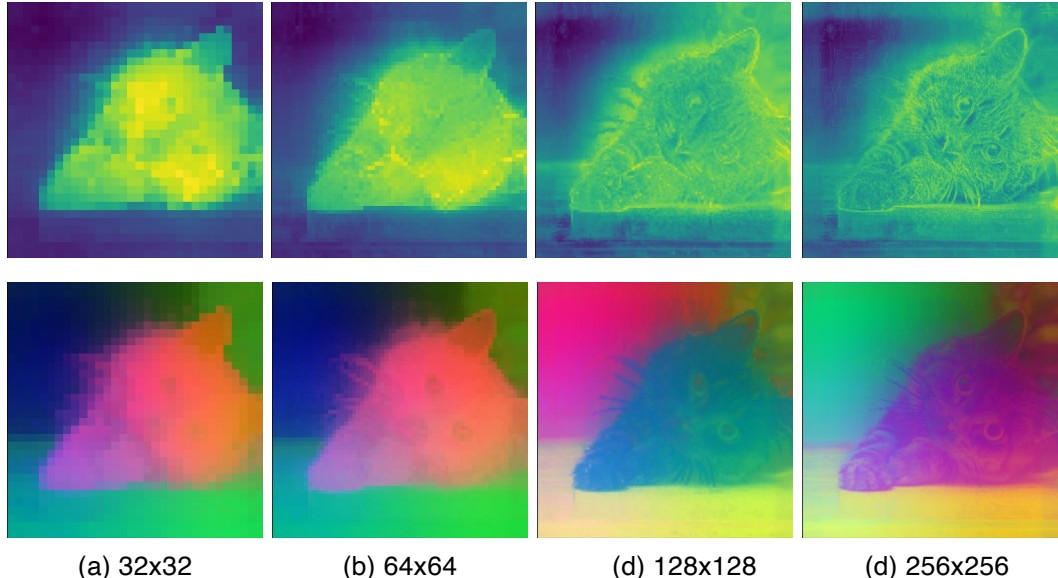

|  (a) 32x32 | (b) 64x64 | (d) 128x128 | (d) 256x256 |

Figure 4: ViT-B/16 student extracts robust features against different input resolutions across $512^2$, $1024^2$, $2048^2$, and $4096^2$. **Top**: cosine map between cls token and patch tokens. **Bottom**: PCA visualization into RGB for features projected to the DINO head.

We visualize the PCA projected feature as well as the cosine similarity map between cls token and patch tokens. As shown in Figure 3, the features from backbone shows satisfactory semantic information, while the cosine map suggests that the cls token fails to capture discriminative information for classification. Furthermore, both the CLIP and MAE head fail to simultaneously capture clean semantic clues together with discriminative information. In contrast, the DINO projected feature captures rich semantic information and presents the best concentration of the cls token. With these obervations, we utilize the model token projected to the DINO head for classification. Analogously, we employ the deep features projected to the DINO head for dense prediction tasks.

