# OpenReview forum: "UNIFIED MULTI-TEACHER DISTILLATION ACROSS HYBRID NEURAL ARCHITECTURES"
_ICLR.cc/2026/Conference — ICLR 2026 Conference Withdrawn Submission_

### Official Review · Reviewer_UKHa · 2025-10-30

**Soundness:** 3
**Presentation:** 2
**Contribution:** 3
**Rating:** 4
**Confidence:** 5

**Summary:**

The authors present a unified framework designed to transfer the knowledge embedded in multiple vision foundation models into both convolutional networks (CNNs) and vision Transformers (ViTs), utilizing training on the ImageNet-1k dataset. They introduce a learnable model token equipped with intra-space and inter-space modules, aimed at exploring visual feature alignment across various representational spaces. This many-to-many projection strategy is straightforward yet effective, yielding consistent downstream improvements and facilitating unified knowledge transfer from pre-trained Transformers, CNNs, as well as their combinations. The experiments demonstrate promising results.

**Strengths:**

1. A clear and modular design that addressed challenging cross-architecture feature alignment.
2. The framework is practical, which is training on an ImageNet-1K scale. Different schemes are designed for CNNs and Transformers.
3. Downstream evaluation shows  the effectiveness of the proposed method, which can be integrated into existing knowledge distillation (KD) pipelines.
4. The design of many-to-many student-to-teacher feature projection and alignment is promising.

**Weaknesses:**

1. While the experimental results are promising, the underlying theory and rationale for these improvements remain largely unexplored and unclear.
2. The procedure, formulation and corresponding modules in Fig. 2 related to cross-space and intra-space interactions are not well described. Thus, they need to be reorganized.
3. Although the proposed method can easily scale with the number of teachers, it lacks analysis and ablation studies regarding this scalability.
4. Details about the projection functions in cross-space and intra-space interactions should be included in the main text rather than relegated to Appendix A.2. Additionally, many implementation details are missing, making it difficult to assess how features are projected effectively. Reproducibility details, such as teacher checkpoints, loss balancing strategies, memory/throughput trade-offs, could be more precise.
5. The token and feature alignment losses have not been formulated clearly. Appendix A.3 only provided brief descriptions. Have all teachers been considered under equal weights?

**Questions:**

1. How might the number of teachers, M, impact the effectiveness of multi-teacher distillation?
2. The primary contribution of this study is the development of multiple feature projections from students to teachers. Given that the features provided by teachers contain richer information and are higher in dimensions, it is generally easier to compress this information rather than expand it. Would there be an equivalent back-projection from teachers to students?

---

### Official Review · Reviewer_oXsB · 2025-10-31

**Soundness:** 2
**Presentation:** 2
**Contribution:** 2
**Rating:** 2
**Confidence:** 4

**Summary:**

This paper studies the problem of multi-teacher distillation and finds existing methods neglect the heterogeneity between teachers. To address the issue, the authors propose a unified framework to transfer the knowledge from multiple vision foundation models into student networks, spanning from CNNs to ViTs. To be specific, the authors introduce a learnable model token that interacts with visual features across multiple representational spaces and find this simple strategy achieves promising empirical results. Extensive experiments on different tasks demonstrate the effectiveness of the proposed method.

**Strengths:**

1. The direction of leveraging multiple vision foundation models as teacher seems interesting and indeed worths exploration. Different vision foundation models have their strength and how to distill them into a single model while keeping their strength is an important problem.
2. It is appreciated that the authors have conduct experiments on different tasks, including ImageNet classification, fine-grained  classification, semantic segmentation and depth estimation.
3. The writing is clear and the paper is easy to follow. The overall structure is well organized and the idea is presented in a coherent manner.

**Weaknesses:**

1. The comparisons with existing methods seems unfair. In Tab.1, the authors tries to compare existing methods with the proposed method which is distilled from the newly released DINOv3. This could be problematic as different methods have different teachers and comparing them side by side in one table is strange.
2. There are limited baselines shown in paper. The proposed approach is only compared with the Proteus and RADIOv2.5 in the hybrid teacher architectures setting, and no other distillation baselines are shown in the paper. Even compare with the teacher network at the same scale, the proposed method does not seem better in most cases which contradicts the statement that 'surpassesall itssourcemodelsindownstreamtransfer performance'.
3. Although the authors claim that the proposed method can transfer the embedded knowledge from foundation models with much smaller dataset, they only conduct experiments on 3 fine-grained classification datasets which is very limited and can hardly examine the generalization ability of the distilled student. Moreover, the fine-grained classification results of Proteus seems to be neglected by the authors and the comparisons should be added.
4. The experimental setup for ablation is unconvincing. The authors use loss value and results on fine-grained classification to verify the efficacy of the design choices which may not ideal as they have huge variances and are not reliable.
5. The dense prediction performance of the proposed method is not favorable as it has a huge gap between its teacher, DINOv3.

**Questions:**

1. It seems that the segmentation results of Proteus is neglected by the authors. Is there a particular reason for this?

---

### Official Review · Reviewer_VL2R · 2025-11-01

**Soundness:** 4
**Presentation:** 1
**Contribution:** 2
**Rating:** 4
**Confidence:** 5

**Summary:**

The paper presents a unified distillation framework to transfer knowledge embedded within multiple vision foundation models into a single CNN or ViT.  The core motivation of the method is that the pre-trained knowledge could be obtained by using distillation. Experiments demonstrate the effectiveness of the method.

**Strengths:**

1. The core motivation of the paper is promising. From my own perspective, knowledge distillation should be an efficient tool to avoid repetitive pre-training. The work in this research line deserves to be noticed by the community.

2. The paper is organized well, which makes it easy to follow.

**Weaknesses:**

1. Insufficient Discussion of Related Work and Overclaimed Contributions

My primary concern is that this paper inadequately discusses highly related work, leading to overclaimed motivations and contributions despite citing relevant papers.

(1) The paper repeatedly emphasizes using only 1/1000 scale of pre-training data as a novel contribution. However, knowledge distillation achieving similar data efficiency is well-established in the literature, yet the paper fails to adequately position itself within this research line.

(2) MobileSAM [1] demonstrated effective distillation using only 1% of training data to create significantly smaller models, with subsequent works [2,3] building upon this foundation. While the authors claim that works like MobileSAM suffer from performance degradation, they fail to distinguish between upstream zero-shot evaluation and downstream fine-tuning performance, which represent fundamentally different assessment paradigms.

(3) The recent ScaleKD [4] work empirically demonstrates that models can acquire teacher knowledge from upstream pre-training via feature distillation using ImageNet-1K as a proxy dataset. This paper neither cites this work nor acknowledges this established phenomenon.

(4) The paper requires a dedicated section discussing developments in this research area. Without proper contextualization, **this manuscript misleadingly suggests it is the first to discover knowledge distillation's data efficiency capabilities**.


2. Concerns about the experimental setups.

(1) Although conducting knowledge distillation from foundation models is promising, why arrange the evaluation on datasets like ImageNet-1K and ADE-20K?

(2) There have been some works that explore conducting knowledge distillation from multiple foundation models, such as MoVE-KD [5], whose motivation is to serve as the vision encoder in MLLM. In this case, different foundational models indeed have a large performance gap on different tasks. As a result, previous MLLM, like Deepseek-VL [6] and NVIDIA Eagle series [7,8], ensembles multiple vision encoders.

(3) My question is, under the setting of this paper, does conducting distillation from multiple teachers urgently need?  Specifically, the extra performance gain from distillation is as clear as that from extra fine-tuning or using more advanced architectures, like in semantic segmentation and depth estimation. If not, why is conducting knowledge distillation under this setting important?


3. Building on the previous concerns and comparing with ScaleKD [4]: why conduct multi-teacher distillation from models of similar parameter scales rather than distilling from a single, stronger teacher? This fundamental design choice lacks adequate justification.

| Method  | Student Arch | Distill Setting        | # imgs | ImageNet | ADE20K |
|---------|--------------|------------------------|--------|----------|--------|
| ScaleKD | ResNet-50    | ViT -> CNN            | 1.2    | 82.03    | 44.5   |
| UMT     | ResNet-50    | Multiple CNN -> CNN   | 1.2    | 78       | 20.6   |
| ScaleKD | ConvNeXt-T   | ViT -> CNN            | 1.2    | 84.1     | -      |
| UMT     | ConvNeXt-B   | Multiple ViT -> CNN   | 1.2    | 84.1     | 34.7   |
| ScaleKD | ViT-B/16     | ViT -> CNN            | 1.2    | 85.5     | 50.8   |
| UMT     | ViT-B/16     | Multiple ViT -> CNN   | 1.2    | 85.6     | 46.5   |




[1] Zhang, Chaoning, et al. "Faster segment anything: Towards lightweight sam for mobile applications." arXiv preprint arXiv:2306.14289, 2023

[2] Xiong, Yunyang, et al. "Efficientsam: Leveraged masked image pretraining for efficient segment anything." In CVPR, 2024

[3] Zhou, Chong, et al. "Edgesam: Prompt-in-the-loop distillation for on-device deployment of sam." arXiv preprint arXiv:2312.06660, 2023

[4] Fan, Jiawei, et al. "Scalekd: Strong vision transformers could be excellent teachers." In NeurIPS, 2024

[5] Cao, Jiajun, et al. "Move-kd: Knowledge distillation for vlms with mixture of visual encoders. In CVPR, 2025.

[6] Lu, Haoyu, et al. "Deepseek-vl: towards real-world vision-language understanding." arXiv preprint arXiv:2403.05525, 2024.

[7] Villa, Andrés, et al. "EAGLE: Enhanced Visual Grounding Minimizes Hallucinations in Instructional Multimodal Models." arXiv preprint arXiv:2501.02699， 2025

[8] Li, Zhiqi, et al. "Eagle 2: Building post-training data strategies from scratch for frontier vision-language models." arXiv preprint arXiv:2501.14818, 2025.

**Questions:**

See Weakness

---

### Official Review · Reviewer_CxKg · 2025-11-04

**Soundness:** 3
**Presentation:** 2
**Contribution:** 3
**Rating:** 6
**Confidence:** 2

**Summary:**

This paper proposes Unified Multi-Teacher Distillation (UMT), a framework for transferring knowledge from multiple heterogeneous vision foundation models (VFMs), such as CLIP, MAE, and DINO-v3, into a single, versatile student model.
Unlike previous multi-teacher approaches (e.g., RADIO, Proteus) that rely on billion-scale datasets and hand-crafted, layer-wise alignment losses, UMT introduces a learnable model token that mediates the interaction between the student and multiple teachers through alternating cross-space and intra-space modules.

Concretely, the student’s features are projected into teacher-specific representational spaces, each paired with the shared model token. The cross-space module allows the token to exchange information across teachers, while the intra-space module reorganizes features within each space, enabling automatic feature alignment without manual loss design.
Training uses ImageNet-1K as a small proxy dataset and simple alignment losses (MSE or cosine similarity) between the model token and teacher features.

Experiments cover four downstream task types: image classification (ImageNet-1K), fine-grained classification (Aircraft, Caltech101, CUB-2011), semantic segmentation (ADE20K), and depth estimation (NYUv2), using both CNN and ViT students.
Results show that UMT achieves performance competitive with, and occasionally superior to, individual teachers and prior multi-teacher baselines, despite training on 1000× fewer samples than RADIO.
Ablation studies highlight the importance of the model token and of the cross-/intra-space interaction modules.
The paper positions UMT as a data-efficient, architecture-agnostic alternative for unifying multiple vision foundation models.

**Strengths:**

The paper introduces a novel perspective on multi-teacher distillation by introducing a learnable model token that serves as a unifying mediator between multiple heterogeneous teachers. This token-based mediation, coupled with alternating cross-space and intra-space interaction modules, provides a creative and architecture-agnostic way to fuse knowledge from diverse foundation models (CNNs and ViTs). Unlike prior approaches (e.g., RADIO, Proteus), UMT eliminates the need for handcrafted, layer-wise or architecture-specific distillation losses, this conceptual simplification is both elegant and original.

The framework is technically sound and systematically evaluated. The authors train students from scratch using ImageNet-1K, yet achieve results competitive with billion-scale pretraining setups. The experiments cover multiple architectures (ResNet, ConvNeXt, ViT-B, ViT-L) and diverse tasks (classification, segmentation, depth estimation), showing robustness and transferability. The ablation studies convincingly isolate the contributions of the model token and interaction modules.

The conceptual diagrams (especially Figures 1 and 2) effectively communicate the architectural flow and the role of the model token. The distinction between cross-space and intra-space interactions is clear and helps readers grasp the novelty. Despite some presentation flaws, the paper’s core idea and motivation are easy to understand and compellingly framed.

**Weaknesses:**

Lack of explicit formulation of the training objective.
Although the paper repeatedly refers to “model-token alignment” and “feature alignment” losses, it never provides a formal equation for the overall distillation objective. The weighting of losses, their aggregation across teachers, and the exact optimization target remain implicit. Appendix A.3 contains only a textual description of MSE and cosine losses.

Quantitative inconsistencies between text and tables.
In Section 4.2, the paper claims a “26.7 % mIoU gain” of UMT over the MAE teacher, but Table 1 reports 46.5 vs 22.2 mIoU (a 24.3 point gain); 26.7 corresponds instead to RADIO-v2.5. This should be corrected or clarified.

Ambiguity in notation and baselines.
Entries like “ViT→RADIO” in Table 1 are confusing, since RADIO is a framework rather than an architecture; readers cannot easily tell whether “RADIO” denotes a student model, a method, or a teacher set.

Missing implementation details.
The paper omits practical details such as the dimensionality of the model token, projection-head architectures Γᵢ, and the number of teachers M used in each experiment. These details are crucial for reproducibility.  Please include a concise equation for the total loss (e.g., a weighted sum of per-teacher token and feature losses) in the main text to make the training setup fully reproducible and mathematically transparent.

Presentation issues and minor errors.
Several typos (“Mutli-teacher”), inconsistent citation formatting (“Oquab et al.” without year), and uneven capitalization (“ImageNet-1k” vs “ImageNet-1K”) detract from polish.

Limited discussion of failure cases and computational cost.
While the method claims data-efficiency, there is no quantitative analysis of computation (FLOPs, training time) or discussion of where UMT fails (e.g., why CNN students underperform).

**Questions:**

See Weaknesses.

---

### Note · Authors · 2025-11-14

I have read and agree with the venue's withdrawal policy on behalf of myself and my co-authors.